# The mRNA Vaccine Expressing Single and Fused Structural Proteins of Porcine Reproductive and Respiratory Syndrome Induces Strong Cellular and Humoral Immune Responses in BalB/C Mice

**DOI:** 10.3390/v16040544

**Published:** 2024-03-30

**Authors:** Luoyi Zhou, Ashenafi Kiros Wubshet, Jiangrong Zhang, Shitong Hou, Kaishen Yao, Qiuyi Zhao, Junfei Dai, Yongsheng Liu, Yaozhong Ding, Jie Zhang, Yuefeng Sun

**Affiliations:** 1College of Animal Science and Technology, Hebei Normal University of Science and Technology, Qinhuangdao 066600, China; zhouluoyi.china@gmail.com (L.Z.);; 2State Key Laboratory for Animal Disease Control and Prevention, Lanzhou Veterinary Research Institute, College of Veterinary Medicine, Lanzhou University, Chinese Academy of Agricultural Sciences, Lanzhou 730000, China; 3Zhaoqing Branch Center of Guangdong Laboratory for Lingnan Modern Agricultural Science and Technology, Zhaoqing 526000, China; yaokaishen.china@gmail.com

**Keywords:** mRNA vaccine, PRRS, PRRSV, structural protein, in vitro transcription, ICS

## Abstract

PRRS is a viral disease that profoundly impacts the global swine industry, causing significant economic losses. The development of a novel and effective vaccine is crucial to halt the rapid transmission of this virus. There have been several vaccination attempts against PRRSV using both traditional and alternative vaccine design development approaches. Unfortunately, there is no currently available vaccine that can completely control this disease. Thus, our study aimed to develop an mRNA vaccine using the antigens expressed by single or fused PRRSV structural proteins. In this study, the nucleotide sequence of the immunogenic mRNA was determined by considering the antigenicity of structural proteins and the stability of spatial structure. Purified GP5 protein served as the detection antigen in the immunological evaluation. Furthermore, cellular mRNA expression was detected by immunofluorescence and western blotting. In a mice experiment, the Ab titer in serum and the activation of spleen lymphocytes triggered by the antigen were detected by ELISA and ICS, respectively. Our findings demonstrated that both mRNA vaccines can significantly stimulate cellular and humoral immune responses. More specifically, the GP5-mRNA exhibited an immunological response that was similar to that of the commercially available vaccine when administered in high doses. To conclude, our vaccine may show promising results against the wild-type virus in a natural host.

## 1. Introduction

Porcine Reproductive and Respiratory Syndrome (PRRS) is a pandemic disease that significantly impacts swine growth. Its primary symptoms manifest as reproductive and respiratory issues across various growth stages in pigs. These include abortion in sows and fetal or premature death in piglets, as well as symptoms like fever, bleeding, and respiratory syndrome. The disease is identifiable by the distinctive blue-purple discoloration of the ears in affected pigs, often referred to as “pig blue ear disease”. Due to its high infectivity and severity, PRRS has been classified as a Class B infectious disease by the World Organization for Animal Health (WOAH) and as a second-class infectious disease in China. Consequently, PRRS has emerged as a major challenge influencing the global pig industry’s development.

PRRSV is a single-stranded positive-sence RNA virus of the genus β-arterivirus. The nucleocapsid of PRRSV has smooth icosahedral symmetry; its diameter is about 30 nm, and it is covered with an envelope. Infectious viral particles are rapidly inactivated by high and low pH (pH < 6.5 or pH > 7.5) and high temperatures (such as exposure to 40 min at 60 °C), but they survive for a long time in a neutral low-temperature environment and are easy to destroy by fat-soluble solvents such as organic solvents [1]. The PRRSV genome is about 15.4 kb and contains at least 10 open reading frames (ORFs), encoding at least 14 non-structural proteins (NSPs) and 8 structural proteins (GP2a, GP2b, GP3, GP4, GP5, 5a, M, and N).

There are mainly two genotypes of PRRSV: PRRSV-1 and PRRSV-2, both of which have shown a global epidemic trend. The main strains prevalent in China are PRRSV-2, among which HP-PRRSV, NADC30-like, and NADC34-like are becoming more and more tripartite in recent years [2,3].

As early as 1994, the United States launched the first commercial PRRS vaccine (a live attenuated vaccine), and China approved the registration of an attenuated vaccine based on isolated CH-1a in 2007 [4,5]. Subsequently, a variety of attenuated and inactivated vaccines were introduced, one after another. Although researchers have been committed to the technological innovation of traditional vaccines, the epidemic situation over the years shows that live attenuated vaccines and inactivated vaccines make it difficult to achieve complete prevention and control of PRRS. The challenges of low immune response to inactivated vaccines and the recombination of attenuated vaccines with field strains, leading to the emergence of new strains, are still unresolved. Therefore, the research on a PRRSV vaccine began to turn to the field of new vaccines such as viral vector vaccines and DNA vaccines [6,7] and achieved certain results, but it is very difficult to eliminate PRRS completely by these vaccines.

The sudden outbreak of COVID-19 rapidly accelerated the research and development of mRNA vaccines, paving the way for the prevention and control of other highly mutated pathogens. The rapid development of mRNA vaccines brings great hope for the complete prevention and control of the PRRS epidemic. In this study, based on the non-replicating mRNA vaccine expression platform, an mRNA vaccine against the Chinese HP-PRRSv strain was obtained, and it showed good immune efficacy. The overall research scheme is depicted in Figure 1.

## 2. Materials and Methods

### 2.1. Ethical Statement

All animal experiments were conducted in accordance with the guidelines laid out in the Guide for the Care and Use of Laboratory Animals by the Lanzhou Veterinary Research Institute (LVRI) of the Chinese Academy of Agricultural Sciences (CAAS, Beijing, China). The Animal Ethics Committee of LVRI (No. SYXK (GAN) 2010-003), CAAS, China, approved all the protocols The approved ethics number is LVRIAEC-2023-063. The Institutional Animal Care and Use Committee (IACUC) of LVRI and CAAS approved the standard protocols.

### 2.2. Cell Culture and Virus

Both the 293T and Marc-145 cell lines were grown and subcultured in high glucose DMEM (JSBio, Seoul, Republic of Korea) with 10% fetal bovine serum (FBS, Gibco, Grand Island, NY, USA) and 1% penicillin-streptomycin (also from Gibco). The HP-PRRSV type (GSWW/CHA 2015) PRRSV strain was cultured in Marc-145 cells. The virus titer was detected by the end point dilution method [8,9] and expressed as the median of tissue culture infection (TCID50/mL).

### 2.3. In Vitro Transcription and Cellular Expression Validation

#### 2.3.1. The Construction of Recombinant Plasmids and In Vitro Transcription

In this study, the plasmid vector W32, derived from the eukaryotic expression plasmid pCDNA3.1, adheres to the original design principles of mRNA vaccines exemplified by COVID-19 [10]. Notably, it encompasses both the eukaryotic promoter CMV and the prokaryotic promoter T7. Utilizing the same vector facilitates both plasmid expression verification and in vitro transcription operations. In consideration of various aspects including the antigenicity of PRRSV structural proteins, the capacity to induce neutralizing antibodies, and the spatial stability of fusion proteins, we comprehensively evaluated these factors. With the assistance of prediction tools such as AlphaFold2, we ultimately identified GP5 and GP2-GP5-M as the two mRNA sequences for this study. We planned to add a Flag tag to the C-terminus of the GP5 and GP2-GP5-M proteins, so the Flag gene sequence was simultaneously constructed into the CDS region of the mRNA. The recombinant plasmid W32 underwent digestion with the restriction endonuclease EcoR I, followed by construction of the recombinant plasmid via homologous recombination (Vazyme). Subsequently, two DNA fragments, PCR-amplified from PRRSV GP5 and PRRSV GP2-GP5-M, were employed to amplify the EGFP DNA and recover the gel. After transformation, the right recombinants were confirmed by double enzyme digestion and sequencing. Thus, three recombinant plasmids—W32-EGFP, W32-GP5, and W32-GP2-GP5-M—were generated.

The plasmids W32-EGFP, W32-GP5, and W32-GP2-GP5-M were linearized using the restriction endonuclease BspQ. Then, gel electrophoresis was used to validate linearization after the nucleic acid had been purified using an alkaline phenol-chloroform-isoamyl alcohol process. 

The preparation of the in vitro transcription system followed the guidelines provided with the mMESSAGE mMACHINE^®^ Kit. Transcription was carried out at 37 °C for 3 h. Following the reaction, DNase was promptly added and incubated at 37 °C for 15 min to degrade any residual DNA template. The reaction volume was increased to 10 times its original volume by adding non-enzymatic sterile water, and the concentration was subsequently determined. mRNA purification was achieved using the acid phenol-chloroform-isoamyl alcohol method. Subsequently, the in vitro transcriptional products were subjected to RNA electrophoresis for analysis. The resulting transcripts were designated as EGFP-mRNA, GP5-mRNA, and GP2-GP5-M-mRNA, respectively. These transcripts were aliquoted into tubes with a quantity of 10 μg per tube and stored at −80 °C.

#### 2.3.2. Cellular Expression Validation

The 293T cells were thawed and poured into a 6-well cell culture in aseptic condition after their state was restored and contamination-free. One vial of each of the two mRNA types was retrieved from −80 °C, thawed on ice, and then incubated overnight in a CO_2_ environment. To transfect the cells, we used LipofectamineTM 3000 Reagent and 2 μg of EGFP-mRNA, GP5-mRNA, and GP2-GP5-M-mRNA was added separately. After incubating at 37 °C in CO_2_ for 6 h, the culture medium was replenished.

At 8, 16-, 24-, 36-, and 48-h post-transfection, the level expression of EGFP-mRNA-in transfected cells were observed using a fluorescence microscope. At 48 h post-transfection, cell samples were collected and subjected to ultrasonication using a RIPA lysis buffer (Beyotime, Shanghai, China) and 1% Protease Inhibitor (PI, NCM Biotech, Suzhou, China). For western blot analysis, a protein sample were thoroughly mixed with 1/3 volume of 4 × Loading Buffer (solarbio, Beijing, China) and denatured at 95 °C for 10 min. The membrane was left to incubate overnight at 4 degrees Celsius with primary antibodies: mouse anti-Flag antibody (BBI Life Sciences, Shanghai, China) and mouse anti-β-actin antibody (Abcam, Cambridge, UK). After three consecutive five-minute washes with a TBST buffer, the membrane was re-incubated with secondary antibodies: (HRP)-conjugated goat anti-rabbit or goat anti-mouse IgG. After three final washes, it was detected by chemiluminescent.

### 2.4. Lipid Nanoparticle Encapsulation

Western blot analysis verified that GP5-mRNA was successfully expressed and that GP2-GP5-M may be expressed, and the encapsulation of liposomes was carried out using the LNP synthesizing apparatus supplied from ENO Biology. We then evaluated the particle size and Zeta potential of GP5-mRNA-LNP and M-mRNA-LNP, each measured three times using the Zetasizer Lab (Malvern Panalytical, Malvern, UK). The Quant-iT™ RiboGreen™ RNA Assay Kit (Invitrogen, Carlsbad, CA, USA) was used for the encapsulation efficiency and concentration of mRNA-LNP.

### 2.5. Animal Immunization and Collection of Serum Samples

#### 2.5.1. Mouse Immunization

A total of 56 eight-week-old Specific-Pathogen-Free (SPF) BalB/c mice were randomly divided into 8 groups as shown in Table 1.

The PRRSV attenuated live vaccine (HP-PRRSV TJM-F92 isolation) was purchased (SinoVET, Beijing, China) in the form of freeze-dried powder. We reconstituted 10 pig doses for 100 mice in 10 mL of PBS for a single immunization. The injection volume was 100 μL per mouse for all six dose groups of mRNA-LNP and the positive control group received the PRRSV attenuated live vaccination.

Each group’s mice received an intramuscular injection of attenuated live vaccination, six doses of mRNA vaccines, and PBS on day 0, and booster injections on day 14, respectively, via intramuscular route, alternating between the left and right hindlimbs. On the second day after injection, mice were observed for any circumstances that could affect the results.

#### 2.5.2. Serum Collection

After around four weeks, 100–200 μL of blood was drawn from the retro-orbital sinus of every mouse using a capillary tube in a microcentrifuge tube. The tube was then left at room temperature for one or two hours. Centrifugation was used at 3000 rpm and −80 °C for 10 min to extract the serum, which was then stored at −80 °C until needed again.

### 2.6. Neutralizing Antibody and Cytokine Detection

#### 2.6.1. Indirect ELISA for Detecting the Titer of Specific Antibodies in Serum

Recombinant PRRSV-GP5 protein (C-His) (Bioss, Beijing, China) was purified and utilized to cover 8 plates in a 96-well enzyme-linked immunosorbent assay (ELISA). Following coating, plates were washed and blocked. Serum was then diluted in gradients of 5 × 4^1^, 5 × 4^2^, 5 × 4^3^, 5 × 4^4^, 5 × 4^5^, and 5 × 4^6^. A total of 50 μL of diluted serum was added to each well in duplicate. Each plate was incubated at 4 °C for 2 h and then washed three times for 5 min each. Goat Anti-Mouse IgG H&L (HRP) (purchased from Abcam, Cambridge, UK) was diluted to a concentration of 1:10,000 and then incubated at room temperature in the dark for 1 h.

Approximately 100 microliters of TMB was added to each well under dark conditions. The plates were incubated at 37 degrees Celsius for 20 min until the coloration appeared. Then, 100 μL of Stop Solution was applied to each well. The OD values of the samples at 450 nm were measured in a microplate reader (SpectraMax Plus, Amesbury, MA, USA). Coating solution, washing solution, blocking solution, TMB, and stop solution were purchased from Dakowi Biotech, Shenzhen, China.

Calculate the titer of antibodies following these steps: (1) Calculate the Average Absorbance of Blank Wells: Measure and record the absorbance values of the blank wells on each plate; Calculate the average absorbance of the blank wells. (2) Select Experimental Wells within Detection Limits: Identify experimental serum wells with absorbance values greater than the average of the blank wells and below the upper limit of the spectrophotometer. (3) Establish the Standard Curve: Plot a standard curve using the dilution factor (X) against the absorbance ratio (Y), where Y is the absorbance of the sample divided by the average absorbance of the blank wells. (4) Linear Fit: Perform linear regression on the standard curve to obtain the equation of the line, usually in the form of Y = mX + b, where m is the slope and b is the intercept. (5) Calculate Dilution Factor for Desired Absorbance: For each sample, substitute the desired absorbance ratio (four times the average blank absorbance) into the equation; Solve for the corresponding dilution factor (X). (6) Determine Antibody Titer: Calculate the reciprocal of the dilution factor to obtain the antibody titer.

#### 2.6.2. Viral Neutralization Experiment

Marc-145 cells were planted at a rate of 1 × 10^4^ cells per well in a 96-well cell culture plate. The plate was incubated at 37 °C in a 5% CO_2_ incubator until the cell density reached 90% to 95%. The neutralization assay involved a total of 42 mRNA vaccine groups, 1 positive control group, and 1 negative control group per plate, along with 1 cell control group per plate. Sera from attenuated live vaccine-immunized mice served as a positive control for the neutralization experiment. Healthy mouse serum served as a negative control, while mock cells were used without any serum. Four replicates were organized for each group.

All sera were heat-treated at 60 °C for 30 min to inactivate them. Subsequently, all sera were serially diluted at (1:2, 1:4, 1:8, 1:16, 1:32, 1:64, 1:128, and 1:256) on a 96-well plate. The PRRSV GSWW2018 virus was diluted in DMEM to a concentration of 100 TCID_50_/50 μL, with an initial TCID_50_ value of 8.33. A diluted viral solution, equal in volume to the serum, was added to every well except for the cell control well. The virus–serum mixture was completely blended and then placed in a 5% CO_2_ incubator at 37 °C for 2 h.

After incubating for 2 h, cells were washed with EMDM and 100 μL of virus–serum combination was added to each well of the 96-well plate. Meanwhile, 200 μL of DMEM was added to the plates and incubated in a 37 °C 5% CO_2_ incubator for 2 h. The virus–serum mixture was removed, washed, and substituted with DMEM. The plate was incubated under the same conditions, and the results were recorded 5 to 7 days after inoculation.

#### 2.6.3. Intracellular Cytokine Staining (ICS) to Detect the Cytokine Content in Splenic T Lymphocytes

After cervical dislocation, Balb/c mice were euthanized and their bodies were immersed in 75% ethanol for 5–10 min in a laminar flow hood. The mice were then skinned, and bone marrow was flushed from pairs of femur and humerus bones into a 50 mL sterile tube. The bone marrow cells were washed, red blood cells were lysed, and bone marrow-derived dendritic cells (BMDCs) were collected aseptically in a biological safety cabinet. The cells were cultured in a 24-well cell culture plate and stimulated with granulocyte-macrophage colony stimulating factor (GM-CSF) and Interleukin-4 (IL-4) to trigger the differentiation. Lipopolysaccharide (LPS) was added to induce maturation. The matured BMDCs were transfected with GP5-mRNA and GP2-GP5-M-mRNA using electroporation. 

Concurrently, splenocytes were collected from the vaccinated mice spleen, filtered through a 200-nylon mesh, centrifuged, red blood cells lysed, and the resulting splenocyte suspension was cleaned and prepared. After mixing the splenic lymphocytes and BMDCs that have been induced at a ratio of 1:5, mixtures from the vaccine groups were seeded in duplicate wells, whereas the splenocytes from positive and negative control groups were in triplicate in a 96-well plate, at a cells density of 1.8 × 10^6^ viable cells per well. The plate was incubated for one hour at room temperature. Subsequently, 0.4 μL of a protein transport inhibitor (500 × concentration) (purchased from Invitrogen, Carlsbad, CA, USA) was added to each well and incubated at 37 °C for a total of 14 h.

Isolated splenocytes from immunized mice were cultured in a 96-well plate and activated with specific immunogens. The cells were dyed with a zombie dye (purchased from BioLegend, San Diego, CA, USA) in low light conditions to assess their viability and integrity. Block non-specific marker with 50 μL of anti-mouse CD16/CD32 mAb (1:24 dilution in flow cytometry buffer) was used at 4 °C for 5 min. Each well was treated with a 50 μL mixture of antibodies targeting cell surface markers such as anti-mouse CD3, anti-mouse CD4, and anti-mouse CD8 antibodies. The cell membrane was fixed and permeabilized with Fixation/Permeabilization solution (BioLegend). Finally, permeabilized cells were incubated with a combination of intracellular antibodies, such as anti-mouse IFN-γ, anti-mouse TNF-α, and anti-mouse IL-4 antibodies. Below are the details of fluorochromes and their corresponding surface marker staining. APC/Fire™ 750 anti-mouse CD3ε antibody labelled with Brilliant Violet 605™ for detecting mouse CD4 cells, and PerCP/Cyanine5.5 labelled antibody for mouse cells. PE/Cyanine7 anti-mouse CD8a antibody against TNF-α, PE/Dazzle™ 594 specific to mouse anti-IFN-γ antibody labelled with phycoerythrin targeting mouse IL-4. All of the antibodies were bought from BioLegend, San Diego, CA, USA.

## 3. Results

### 3.1. Sequence Design and Plasmid Construction

The main purpose of this study is to develop an mRNA vaccine targeting Highly Pathogenic Porcine Reproductive and Respiratory Syndrome Virus (HP-PRRSV). Extensive research indicates that non-replicating mRNA vaccines have played a role as candidate vaccines in diseases such as HIV, rabies, Zika virus infection, and influenza [11]. However, whether they can be used for the development of pig vaccines remains to be explored [8]. To address this issue, we first constructed three recombinant plasmids based on W32 (Figure 2), where one plasmid is designed to express enhanced green fluorescent protein (EGFP), and the other two carry PRRSV GP5 and PRRSV GP2-GP5-M concatenated sequences, respectively, to express PRRSV-related antigenic proteins. 

For the selection of antigens in the PRRSV-mRNA vaccine, this study primarily relies on predictions related to the antigenicity and stability of structural proteins. GP5 is the most crucial neutralizing antigen in PRRSV’s structural proteins and serves as the main target antigen in this study. Based on this, docking simulations were conducted between the five structural proteins of PRRSV and complexes with the remaining four proteins. The Root Mean Square Deviation (RMSD) values were used to assess the impact of each structural protein on the overall spatial stability. This preliminary analysis excluded GP3 and GP4 as components of the concatenated sequence mRNA vaccine, as shown in Figure 3A. Subsequently, fusion analysis of GP2, GP5, and M proteins revealed high spatial stability in the expression of these three structural proteins. In conclusion, considering the neutralizing ability, antigenicity, and spatial stability of structural proteins, this study ultimately chose the full GP5 sequence (ORF5) and GP2-GP5-M (full ORF sequence connected by GS linkers) as the two components of the mRNA vaccine, the spatial configuration of them can be seen in Figure 3B,C.

### 3.2. In Vitro Transcription and Cellular Expression Validation

#### 3.2.1. In Vitro Transcription

Following the completion of in vitro transcription (IVT), DNA templates were promptly removed using DNase, and the concentration of IVT-mRNA was determined. Subsequently, IVT-mRNA was purified using the phenol-chloroform-isopropanol method, and purity and concentration were reassessed. The calculated purification recovery rate was approximately 50%. The results of in vitro transcription were verified through RNA electrophoresis, as depicted in Figure 4.

#### 3.2.2. Validation of EGFP-mRNA Expression

To demonstrate the rationality and viability of the designed and constructed mRNA expression system and confirm the expression capability of 293T cells for mRNA, expression validation was initially performed using the positive control EGFP-mRNA. This validation employed a visually intuitive method using fluorescence microscopy. In this instance, 293T cells at a density of 80% to 90% were transfected with 2 μg of EGFP-mRNA. The cells were observed and recorded for fluorescence at 8 h, 16 h, 24 h, 36 h, and 48 h post-transfection using fluorescence microscopy. As shown in Figure 5, significant EGFP expression was observed as early as 8 h post-transfection, and it demonstrated a noticeable increasing trend in the subsequent time points. However, after 24 h, as the cell status declined, the expression of EGFP-mRNA gradually decreased. The expression validation for EGFP-mRNA confirmed that the mRNA expression system constructed in this study is highly efficient and functional and capable of expressing in 293T cells.

#### 3.2.3. Expression Validation of GP5-mRNA and GP2-GP5-M-mRNA

After transfecting GP5-mRNA and GP2-GP5-M-mRNA into 293T cells for 48 h, cell samples were collected for western blot analysis. The sizes of the proteins expressed by GP5-mRNA and GP2-GP5-M mRNA are approximately 25 kDa and 73 kDa, respectively. The results indicated that the protein expressed by GP5-mRNA could be detected by an anti-Flag antibody and had the correct size. However, the anti-Flag antibody fails to recognize proteins expressed by GP2-GP5-M-mRNA, as shown in Figure 6. It is hypothesized that the Flag tag fused to the expressed protein may be enclosed internally, preventing it from binding with the antibody.

### 3.3. Lipid Nanoparticle Encapsulation

The encapsulation of mRNA vaccines in lipid nanoparticles (LNPs) is crucial for their in vivo delivery and immune functionality. The results of LNP encapsulation significantly impact the subsequent functions of mRNA vaccines. mRNA-LNP must meet requirements in terms of stability, uniformity, and charge. Therefore, it is necessary to assess parameters such as particle size, Zeta potential, Polydispersity Index (PDI), encapsulation efficiency, and encapsulation concentration. As shown in Table 2, the encapsulation results for both GP5-mRNA and GP2-GP5-M-mRNA meet the required standards. Figure 7 shows the peaks of particle size and zeta potential.

### 3.4. Detection of GP5-Specific Antibody Titers Using the Indirect ELISA

After the mRNA expressing the corresponding protein enters the body, it is recognized and taken up by antigen-presenting cells, entering the cytoplasm and being translated into the respective protein by ribosomes. This protein is either degraded into smaller fragments (peptides) by proteasomes or transported to the extracellular space through the Golgi apparatus. The released protein fragments in the extracellular environment can be engulfed and degraded into smaller peptide fragments by various immune cells through phagocytosis. These peptide fragments form complexes with MHC class II proteins, expressing the antigen-presenting cells’ surface, recognized by CD4+ T cells, and promoting B cells to produce antigen-specific antibodies. In this study, the indirect ELISA method was employed to measure the titer levels of the antibodies produced in the mouse serum. PRRSV-GP5 protein was coated, and immunized mouse sera with different dilutions were added to detect the levels of specific antibodies produced in the mouse serum.

In Figure 8A,B, we present the OD_450nm_ values obtained from samples of each experimental group and the control group at each dilution factor in the form of line graphs. It can be clearly seen that the results of the GP5-mRNA 10 μg group and the GP5-mRNA 15 μg group are significantly higher than those of the other groups.

After calculating the titers of each sample, it is evident that the titers of the mRNA vaccine group and the positive control group were significantly higher than those of the negative control group. Furthermore, the GP5-specific antibody titers induced by the GP5-mRNA 15μg group were the highest (Figure 8C).

The ELISA results, as shown in Figure 8, indicate that the specific antibody titers in the six mRNA vaccine dose groups and the positive control group were significantly higher than those in the negative control group. Additionally, with the exception of the GP5-mRNA 15 μg group, which induced significantly higher titers of GP5-specific antibodies compared to the positive control group, the induction results in the other mRNA vaccine groups were similar to those in the positive control group.

### 3.5. Viral Neutralization Experiment

The neutralization assay can be conducted by observing the infection of sensitive cells after co-incubation with the serum from immunized mice and the virus. The ability of mRNA vaccines to induce virus-neutralizing antibodies can be determined. During the neutralization experiments using mouse serum from the negative control group (PBS group), no viral neutralization phenomenon was observed. Using the Reed-Muench method, the serum dilution factors at which 50% of the cell wells are protected from lesions were calculated for the six mRNA vaccine groups and the positive control group. This dilution factor represents the neutralizing antibody potency of the serum. As shown in Figure 9, using the RM one-way ANOVA analysis method for significant difference assessment, it was observed that although the high values of the positive control attenuated vaccine group were relatively elevated, there was substantial within-group variance. In comparison to the GP5-mRNA 15 μg group, there was no significant difference observed in the neutralizing capacity against serum. However, the neutralization titers of all other groups were notably weaker than those of the positive control group.

### 3.6. Detection of the Cytokine Content in Splenic T Lymphocytes by Intracellular Cytokine Staining

ICS relies on flow cytometry, utilizing specific antibodies to label intracellular factors, indirectly detecting the expression levels of these factors within cells. This method offers high sensitivity and specificity, enabling the simultaneous detection of multiple cytokines. In this experiment, lymphocytes from the spleen of mice collected 28 days post-immunization were subjected to surface marker labeling and intracellular cytokine staining. The proportions of CD4+ T lymphocytes and CD8+ T lymphocytes expressing IFN-γ, TNF-α, and IL-4 were determined. This analysis helps assess the activation of T lymphocytes in mice after mRNA vaccination, providing insights into the vaccine’s impact on the activation of the cellular immune response.

The results (Figure 10) indicate that, in CD4+ T cells, there is a significant difference in TNF-α secretion among different treatment groups. The proportion of cells secreting TNF-α in the mRNA vaccine groups and the positive control group is significantly higher than that in the negative control group. Additionally, the GP5-mRNA 15 μg group is significantly higher than the positive control group. In CD8+ T cells, the GP5-mRNA 15 μg group demonstrates good performance in the secretion of three cytokines (IFN-γ, TNF-α, and IL-4). It exhibits a significant advantage compared to the positive control group.

## 4. Discussion

The COVID-19 pandemic has accelerated the development of mRNA vaccines, establishing them as a sophisticated and promising vaccine technology. Compared to traditional inactivated and attenuated vaccines currently used for PRRS control, mRNA vaccines offer shorter preparation cycles, extremely high safety, and long-lasting efficacy, holding great promise for complete PRRS prevention and control. However, non-replicating mRNA vaccines have not yet been utilized in PRRS vaccine development, making research and development of PRRS mRNA vaccines a novel field.

In this research, we first constructed a W32-EGFP plasmid containing a complete mRNA transcription system (including 5′-UTR, 3′-UTR, and Poly(A) tail). Subsequently, EGFP-mRNA was synthesized via in vitro transcription and transfected into eukaryotic cells. Finally, significant fluorescence expression was observed under a fluorescence microscope, demonstrating rapid and effective expression of EGFP-mRNA based on this mRNA expression system in cells through in vitro transcription. This confirms the feasibility and efficiency of the mRNA expression system we constructed, as well as the in vitro transcription method used.

In addition to selecting and designing the mRNA sequence encoding the most critical neutralizing antigen GP5 protein from PRRSV as a standalone mRNA, we utilized Alphafold 2 to predict the protein structures and the degree of spatial structural changes after fusion expression. Consequently, we decided to use a GS linker-linked GP2-GP5-M fusion as another mRNA. This mRNA not only incorporates both GP5 and M proteins, which are two important antigenic proteins, but also includes GP2 to stabilize the structure of the fusion protein. To ensure the smooth progression of subsequent mouse experiments, we performed murine optimization on the designed sequences.

Unfortunately, in the validation of cellular expression of the GP2-GP5-M fusion protein at the cellular level, we did not succeed in demonstrating the expression of the fusion protein. However, because nucleic acid gel electrophoresis confirmed the integrity and correct length of the transcribed mRNA product, and considering the possibility of the Flag tag being folded within the fusion protein, we continued to investigate this mRNA further.

This study involved extensive mouse experiments, which included determining the titers of GP5 antibodies in serum using an indirect ELISA and conducting virus neutralization assays to assess the ability of serum to neutralize the virus. The results are described in Figure 8 and Figure 9. In Figure 8A,B, the line graphs illustrate that the OD values of mouse serum in the GP2-GP5-M-mRNA 15 μg group, GP5-mRNA 10 μg group, and GP5-mRNA 15 μg group were generally higher compared to the other groups, including the positive control group, across different dilution factors. Figure 8C presents a linear fit of the OD values of the samples at different dilutions. Utilizing the intercept and slope of the linear equation, the unique titers of each sample were calculated when the OD value exceeded four times that of the blank well. This set of bar graphs clearly shows that the GP5 antibody titer in the serum of the GP5-mRNA 15 μg group was the highest, and the titers of the six mRNA vaccine groups and the positive control group were significantly higher than those of the negative control group. In Figure 9, the results of virus neutralization assays using mouse serum indicate that the neutralization titers of serum from the three GP5-mRNA vaccine dose groups were significantly higher than those of the three GP2-GP5-M-mRNA vaccine dose groups. However, the neutralization abilities of serum from all six mRNA vaccine groups were inferior to that of the positive control group, with only the neutralization titer of serum from the GP5-mRNA 15 μg group showing no significant difference compared to the positive control group (as analyzed by RM one-way ANOVA).

In Figure 8, it is observed that the titers of GP5 antibodies induced by serum from some mRNA vaccine groups are higher than those induced by serum from the positive control group. However, Figure 9 demonstrates that the neutralization ability of serum from the positive control group is significantly stronger than that of serum from five out of the six mRNA vaccine groups, except for the GP5-mRNA 15 μg group. This result may suggest that the antibodies induced by mRNA vaccines are not all neutralizing antibodies or may have lost their neutralizing activity before detection. However, the serum from the GP5-mRNA 15 μg group not only induces high levels of GP5 antibodies but also exhibits neutralization ability comparable to that of the positive control serum, indicating that increasing the dosage may potentially enhance the immunogenicity of mRNA vaccines.

For assessing the ability of mRNA vaccines to induce cellular immunity, this study conducted an assay on mouse splenocytes. We utilized red blood cell lysis, Zombie dyes for distinguishing live lymphocytes, and specific antibodies against immune cell surface markers CD3, CD4, and CD8 to differentiate CD4+ T cells and CD8+ T cells. Subsequently, we calculated the proportions of cells secreting IFN-γ, TNF-α, and IL-4 within these T cell populations. From Figure 10, it can be observed that the overall effectiveness of the GP5-mRNA vaccine group is higher than that of the GP2-GP5-M-mRNA vaccine group. Notably, the GP5-mRNA 15 μg group stands out, inducing higher levels of IFN-γ, TNF-α, and IL-4 secretion by CD8+ T lymphocytes, resulting in a stronger cellular immune response. In the meantime, the activation of CD4+ T lymphocytes in the GP5-mRNA 15 μg group, primarily evidenced by TNF-α production, showed less pronounced induction of IFN-γ and IL-4 secretion, suggesting lower activation levels of Th1 and Th2 cells. Overall, the ability of the mRNA vaccine groups to induce cellular immunity is not weaker than that of the positive control group, with the GP5-mRNA 15 μg vaccine group even surpassing the positive control group in certain aspects.

From the overall results, it is evident that the levels of both humoral and cellular immunity induced by the GP5-mRNA vaccine group are significantly higher than those induced by the GP2-GP5-M-mRNA vaccine group. This finding contradicts the conclusion that the formation of GP5-M protein dimers in naturally occurring viruses triggers host immune responses together. Through calculations and analysis, potential reasons for this inconsistency can be summarized as follows: (1) Under the same vaccine dosage, the GP2-GP5-M-mRNA vaccine may have relatively fewer effective T cell epitopes and B cell epitopes. (2) While fusion proteins promote the stability of their own structure, they may fold and cover important lymphocyte epitopes. (3) The presence of GP2 may hinder the formation of virus-like particles (VLPs), thereby reducing the ability of fusion proteins to induce immune responses. Expressing fusion proteins ensures that various structural proteins can simultaneously enter a single cell, avoiding issues such as the assembly failure of VLPs due to the absence of essential components. However, this approach also exhibits several drawbacks. In subsequent in-depth development, it may be worth considering removing the GP2 component and retaining only GP5 and M as the main components of the mRNA. Additionally, utilizing genetically cleavable linkers within cells could replace the currently used GS linkers.

Furthermore, compared to animals vaccinated with attenuated vaccines, which exhibited slowed growth and displayed aggressive behavior such as biting each other, mice vaccinated with mRNA vaccines did not show any abnormal behaviors such as aggression, slowed growth, or mortality. This suggests that the side effects of mRNA vaccines may be lower than those of attenuated vaccines.

This study also has some limitations. When identifying the expression of GP5-mRNA and GP2-GP5-M-mRNA using western blot analysis, we only used anti-Flag antibodies for detection. This decision was made because the GP5 monoclonal antibody we prepared did not exhibit the expected capability in detecting positive samples. Additionally, this study did not collect serum samples from mice at appropriate time points, which prevents us from elucidating the changes in immune responses induced by mRNA vaccines and attenuated vaccines at various stages. Furthermore, we are unable to determine whether mRNA vaccines can induce the production of neutralizing antibodies more rapidly than attenuated vaccines. Although the overall experimental results indicate that the GP5-mRNA 15 μg group elicited the strongest immune response, we are not certain if this dosage is the most suitable high value.

Certainly, the major limitation of this study lies in the fact that it only involved mouse animal experiments, without conducting immunization and challenge protection experiments in pigs. From published studies, it is understood that PRRSV cannot achieve high levels of replication in rodents such as mice [15]. Although this study focused on mRNA vaccines based on components of PRRSV structural proteins, which do not involve the growth and replication of the complete virus in the host, it still cannot completely determine whether the immune response induced by mRNA vaccines in mice is consistent with that in pigs. In future investigations, it is essential to supplement immunization and challenge protection experiments with mRNA vaccines in pigs.

In conclusion, this study has developed an efficient and safe GP5-mRNA vaccine (15 μg) capable of eliciting high levels of neutralizing antibodies and demonstrating significant activation of cellular immunity. This study explored GP2-GP5-M-mRNA, another mRNA vaccine. Although it did not show significant immune response induction, it offered valuable insights into the possible problems and directions for developing mRNA vaccines that express fusion proteins, which could lead to new opportunities for research in the future. As the PRRS epidemic continues to worsen, mRNA vaccines may be able to help contain or eradicate the disease. Thus, our study has established a strong basis for the investigation of PRRS mRNA vaccines and will facilitate the prevention and management of PRRS on a worldwide scale.

## Figures and Tables

**Figure 1 viruses-16-00544-f001:**
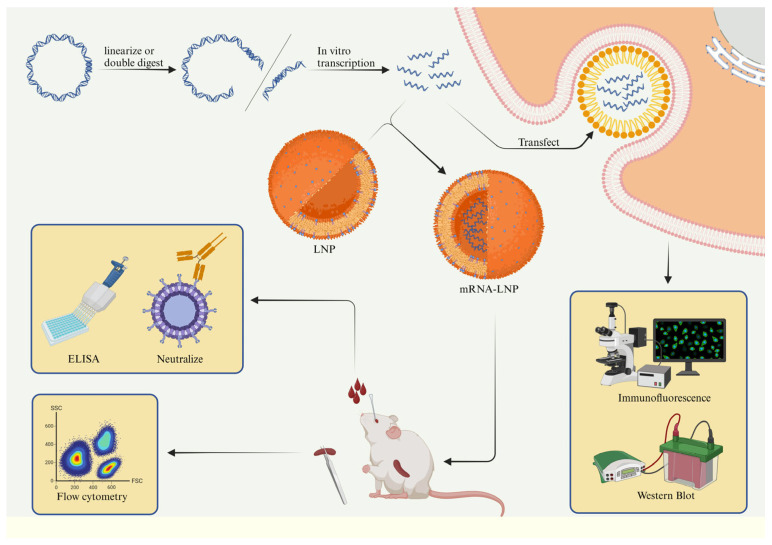
Overall process. Firstly, it is necessary to construct a plasmid carrying the mRNA, purify it, and transcribe it in vitro. Then, transfect the mRNA into 293T cells, and detect the expression by western blot and fluorescence microscopy. Subsequently, the mRNA is encapsulated with lipid nanoparticles (LNPs) to form a complete mRNA vaccine. We immunized mice with the mRNA vaccine and completed the immune evaluation of the mRNA vaccine through ICS (Intracellular Cytokine Staining), ELISA (Enzyme-Linked Immunosorbent Assay), and neutralization experiments.

**Figure 2 viruses-16-00544-f002:**
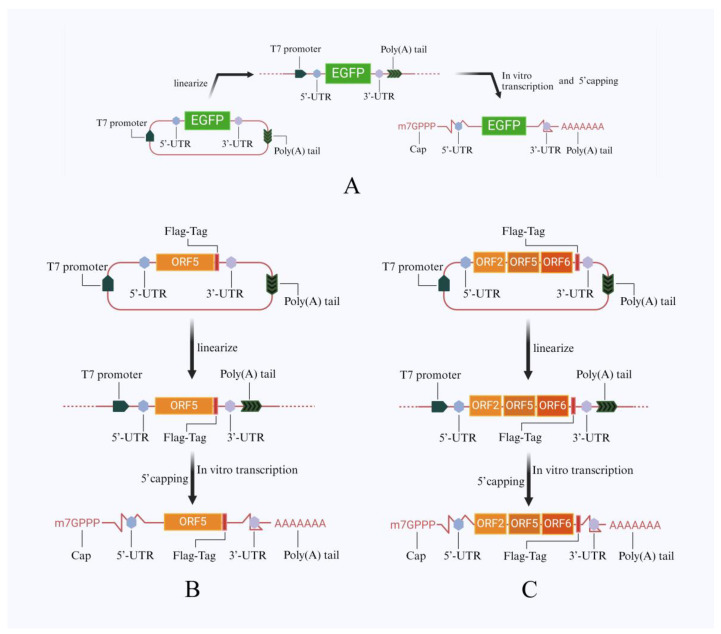
Transcription pathway (Plasmid to mRNA) (**A**–**C**): the schematic diagram respectively depicts the process of EGFP, GP5, and GP2-GP5-M linearization of circular plasmids followed by the completion of in vitro transcription and modification.

**Figure 3 viruses-16-00544-f003:**
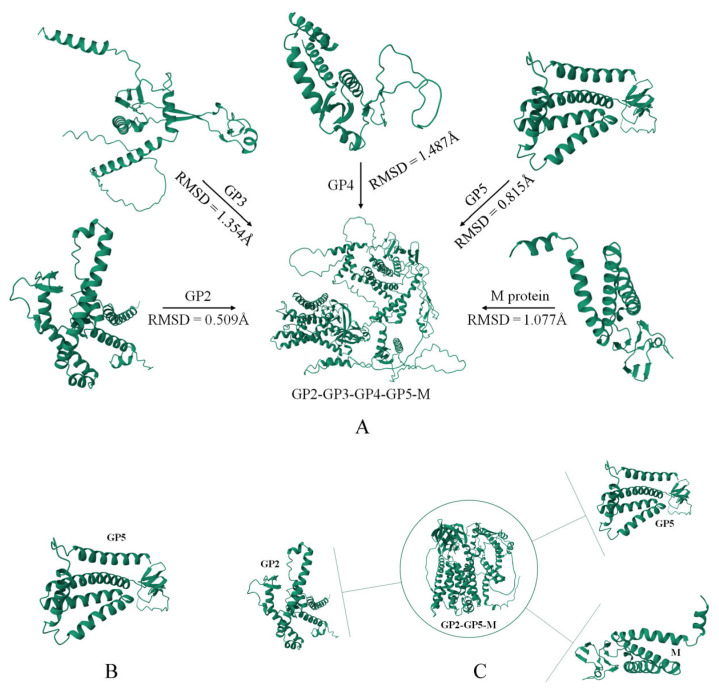
Homology model of PRRSV: The tertiary structure of PRRSV and the antigenic proteins produced by mRNA vaccines were predicted using Alphafold2. (**A**) illustrates the degree of spatial structural changes for five main structural proteins in both independent and complex states, represented by the RMSD values, where a higher value indicates a greater degree of spatial structural alteration. (**B**,**C**) present the spatial structural prediction results for two candidate mRNA sequences.

**Figure 4 viruses-16-00544-f004:**
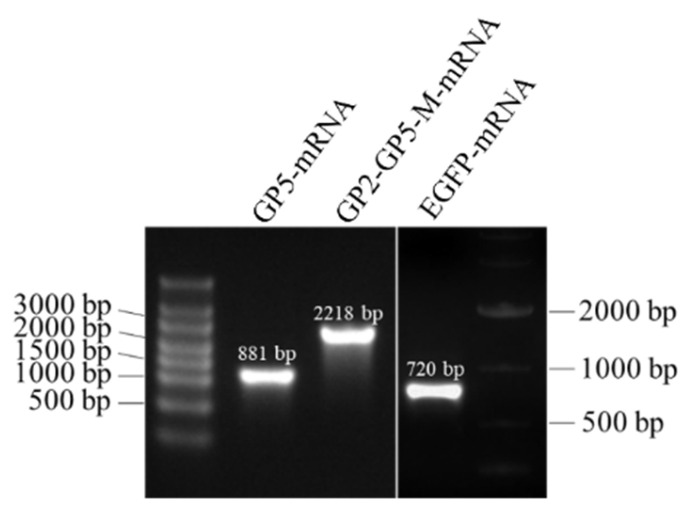
This shows an in vitro transcription.

**Figure 5 viruses-16-00544-f005:**
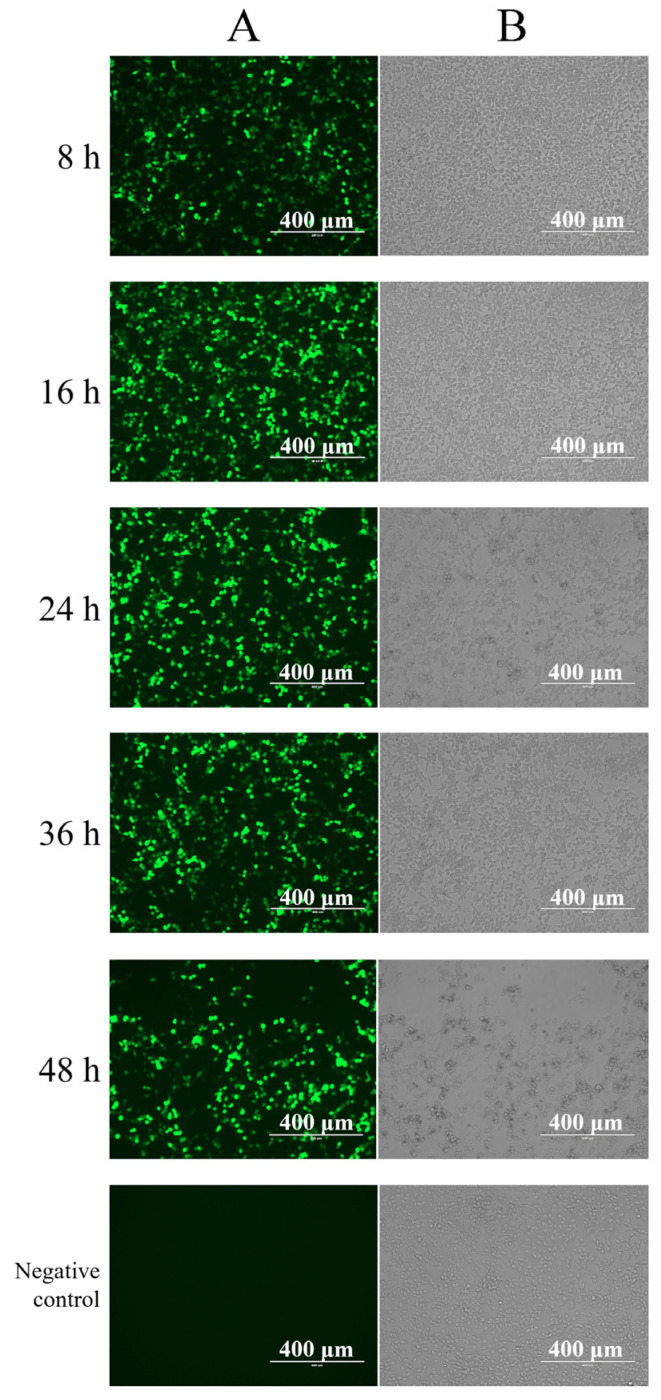
Fluorescence expression of EGFP protein. All components of column (**A**) are images seen under a fluorescence microscope, and all components of column (**B**) are images seen in bright field under the same field of view.

**Figure 6 viruses-16-00544-f006:**
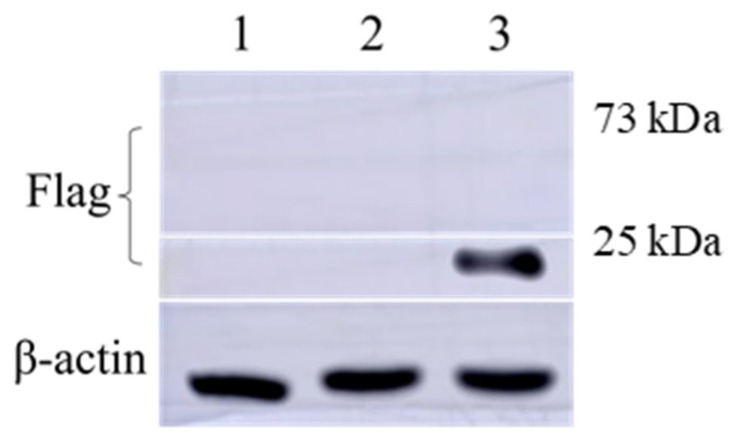
Western blot result of mRNA vaccine. 1 Negative control; 2 GP2-GP5-M-mRNA; 3 GP5-mRNA.

**Figure 7 viruses-16-00544-f007:**
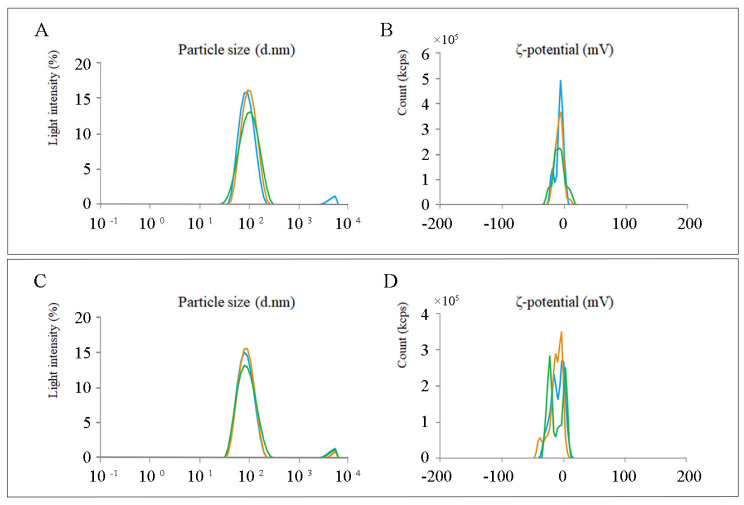
Peak profile of particle size and Zeta potential. (**A**,**C**) represent the particle diameter after encapsulation of GP5-mRNA-LNP and GP2-GP5-M-mRNA-LNP, respectively. This measurement reflects the size and uniformity of lipid nanoparticles after encapsulation, with three replicates for each product. (**B**,**D**) represent the Zeta potential after encapsulation for both cases, reflecting the surface charge of the particles. This information is crucial for assessing whether the products can efficiently enter cells, with three replicates for each group, shown as three peaks in different colors. All analyses were performed using the MALVERN ZSU3100 analyzer.

**Figure 8 viruses-16-00544-f008:**
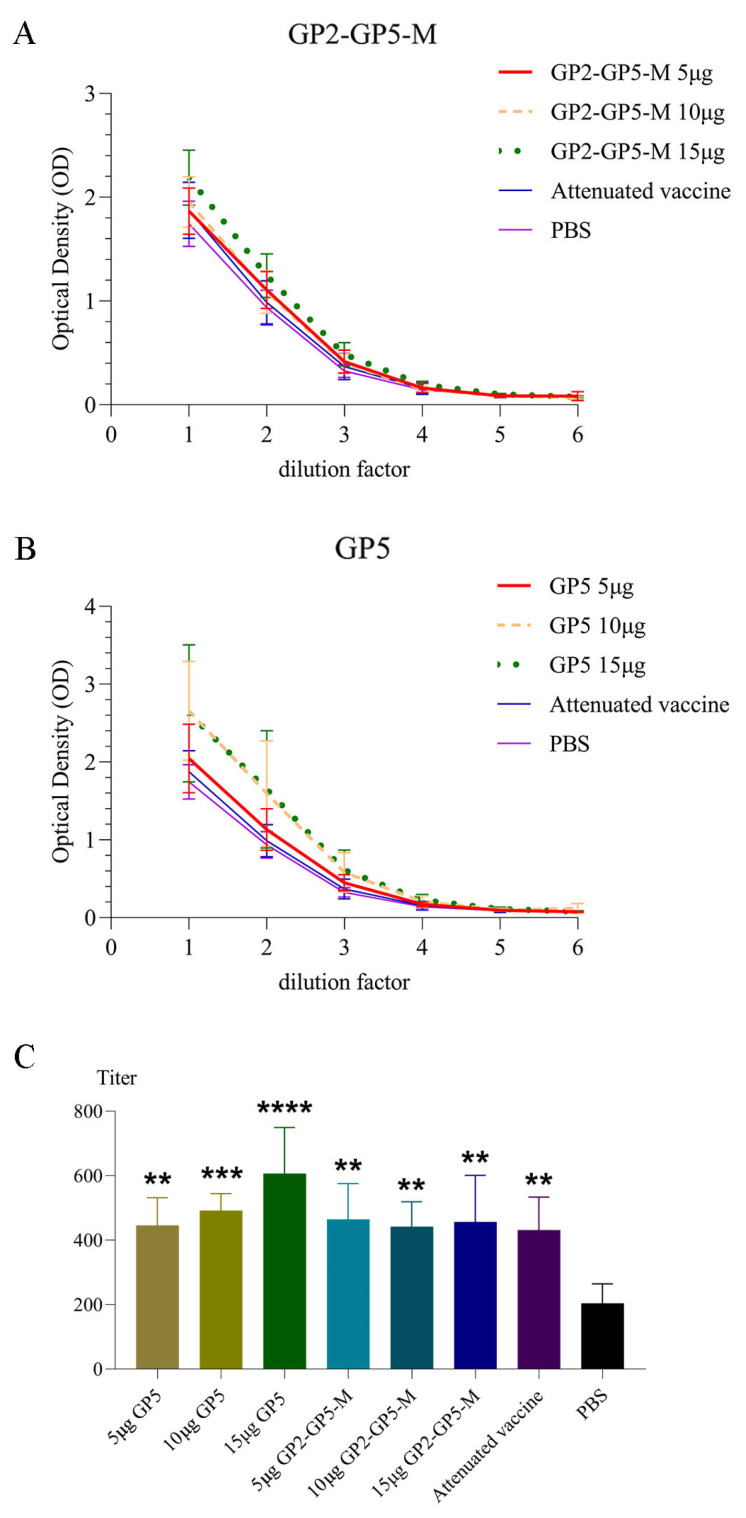
GP5-specific ELISA tiers in serum. In (**A**,**B**), the ordinate represents OD values, while the abscissa represents dilution factors. In the abscissa, numbers 1 to 6 respectively represent dilutions ranging from 5 × 41-fold to 5 × 46-fold. The data of the attenuated vaccine group and the PBS group in the two figures are the same. Through these two line graphs, it is evident that both the GP5-mRNA 10 μg and the GP5-mRNA 15 μg groups induced higher levels of GP5 antibodies. Additionally, the ability of the GP5-mRNA group to induce GP5 antibodies at the same dose is significantly higher than that of the GP2-GP5-M-mRNA group. In (**C**), we utilized the one-way analysis of variance (ANOVA) methodology to conduct an investigation into differences among groups. ** *p* < 0.005; *** *p* < 0.001; **** *p* < 0.0001. The detection of specific antibody titers in mouse serum was conducted using the ELISA method. The results shown in the graph were obtained after processing through a specific calculation method. All vaccine groups exhibited significantly higher results compared to the negative control group.

**Figure 9 viruses-16-00544-f009:**
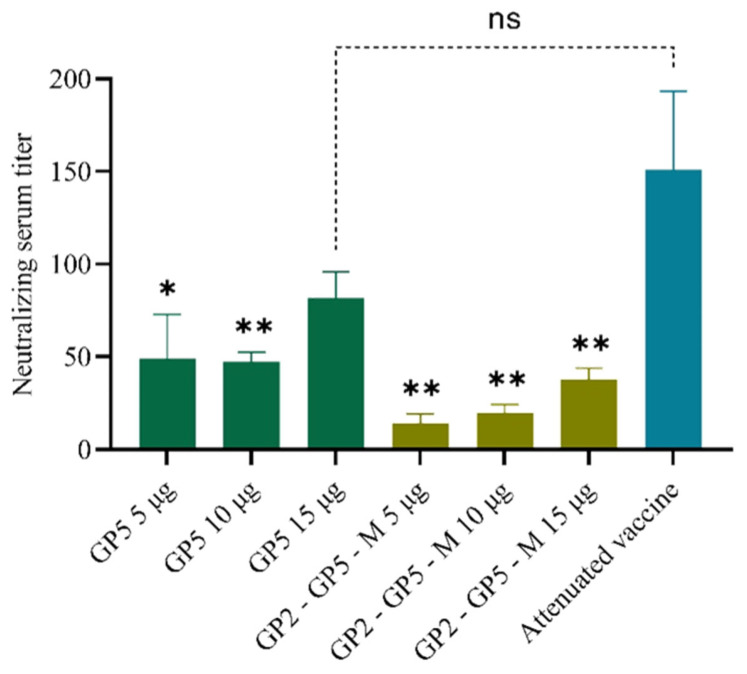
Neutralizing serum titer in mice after immunizing mRNA vaccine. Typically, titers higher than 50 are considered to potentially possess neutralizing activity. All three dosage groups of GP5-mRNA vaccine exhibited serum titers higher than 50, indicating neutralizing activity. However, the serum neutralizing titers were lower in all three dosage groups of the GP2-GP5-M-mRNA vaccine. * *p* < 0.05; ** *p* < 0.005.

**Figure 10 viruses-16-00544-f010:**
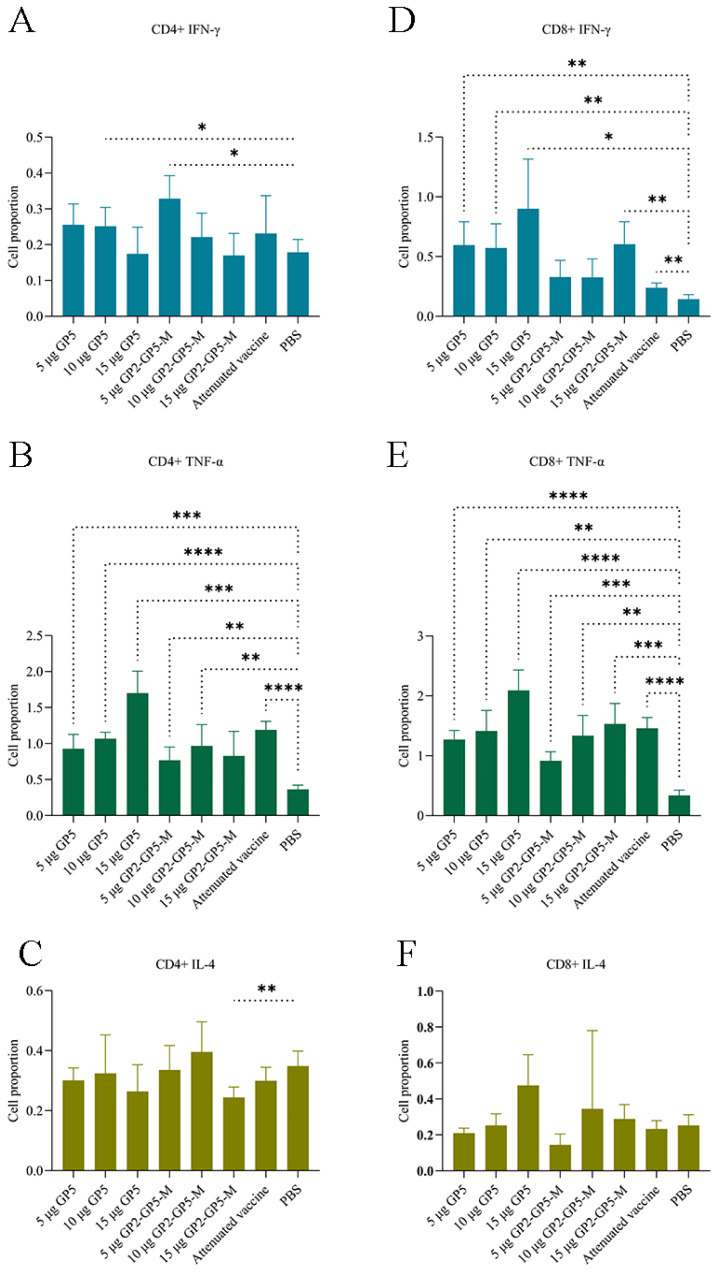
ICS test results of three cytokines. A-C showed the proportion of cells secreting IFN-γ, TNF-α and IL-4 in CD4+ T lymphocytes, and D-F showed the proportion of cells secreting IFN-γ, TNF-α and IL-4 in CD8+ T lymphocytes, respectively. This displays a bar chart derived from the analysis of all data. It reflects the changes in the levels of secretion of three cytokines in cellular immunity induced by mRNA vaccines and the control group. The RM one-way ANOVA method was utilized for assessing the significance of differences, * *p* < 0.05; ** *p* < 0.005; *** *p* < 0.001; **** *p* < 0.0001. Groups that showed no significant differences compared to the negative control group did not demonstrate their level of variability.

**Table 1 viruses-16-00544-t001:** Shows details of the experimental design.

Groups	Type and Dose of Immunogen		
GP5-mRNA groups	5 μg	10 μg	15 μg
GP2-GP5-M-mRNA groups	5 μg	10 μg	15 μg
Positive control group	Immunization with PRRSV attenuated live vaccine		
Negative control group	Inoculation with PBS		

**Table 2 viruses-16-00544-t002:** Quality assessment of mRNA-LNP encapsulation post-packaging [12,13,14].

Inspection Items	Methods	Acceptance Criteria	GP5	GP2-GP5-M
Particle size	DLS	85~150 nm	92.2 nm	86.0 nm
Zeta potential	ELS	≥−10 mV	−7.5 mV	−9.3 mV
PDI	DLS	≤0.250	0.162	0.218
Encapsulation Efficiency	RiboGreen kit	≥90%	93.1%	93.7%
Encapsulation Concentration	RiboGreen kit	≥0.100 mg/mL	0.200 mg/mL	0.200 mg/mL

## Data Availability

During the period of this study, no new databases were established to store experimental data.

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
