# Peer review of "The mRNA Vaccine Expressing Single and Fused Structural Proteins of Porcine Reproductive and Respiratory Syndrome Induces Strong Cellular and Humoral Immune Responses in BalB/C Mice"

_viruses, 2024, doi:10.3390/v16040544_

Round 1

Reviewer 1 Report

Comments and Suggestions for Authors

The paper titled "The mRNA vaccine expressing single and fused structural proteins of PRRSV induces significant cellular and humoral immune responses in BalB/C mice" highlights the emergence of mRNA vaccines as a mature and promising platform for PRRSV vaccine development. In the study, the authors use Alphafold 2 to predict protein structures, demonstrating the feasibility and efficiency of the mRNA expression system. The results reveal that the GP5-mRNA vaccine group, especially at the 15 μg dosage, outperforms the GP2-GP5-M-mRNA group, eliciting a certain level of GP5-specific antibodies and a potent neutralizing effect. This group exhibits a clear cellular immune response, with increased secretion of IFN-γ, TNF-α, and IL-4 in CD8+ T lymphocytes. CD4+ T lymphocytes primarily produce TNF-α, suggesting potential modulation of Th1 and Th2 cells. Significantly, mice vaccinated with mRNA vaccines exhibit no growth hindrance or abnormal behavior, indicating lower side effects than attenuated vaccines.

However, the paper only establishes the proof of concept for the mRNA vaccine against PRRSV, and further evidence is required to assess its efficacy. Based on current results, there is no apparent advantage, or even equal potency, compared to attenuated vaccines. Additionally, understanding the better performance of the GP5 mRNA vaccine group in terms of antibody and neutralizing activity compared to the fused GP5 with GP2 and M is challenging, as the expression of GP2/5 and M proteins is not demonstrated, with only tag antibodies used for Western blot analysis. The use of GFP mRNA transfection for evaluating mRNA vaccine protein expression levels is inappropriate. The paper should include additional fluorescence staining with viral protein-specific antibodies to identify expression levels. A Western blot to confirm the correct expression and an additional control group with an empty vector should be included. Serum collection at more time points should be included to evaluate the dynamic change of antibody level to come up with a more sound conclusion.

Minor Comments:

The abstract is a general introduction. The abstract should summarize the entire manuscript, emphasizing key findings, including experiment design, results, and conclusions.

Line 71-74: a difficult sentence to follow.

Line 78 has a citation issue, the author's name is incorrect, and the mention of replicon vaccine is incorrect.

Figure 1 legend is deemed inappropriate and requires correction.

Line 110 should mention gel electrophoresis instead DNA electrophoresis.

Line 121, a vector control should be included in cellular expression validation.

Figure 7, A B C D in Figure is unclear.

Figure 8 indicates GP5 alone induced higher antibody levels compared to the GP5/2/m fusion construct, a detailed analysis should be provided to elucidate why the combination GP5/2/M construct exhibits even lower neutralizing antibodies in comparison to the GP5 single construct and the figure legend is incorrect, it does not depict neutralizing antibody titers.

Figure 9 indicates significantly lower levels of neutralizing antibodies compared to the attenuated vaccine. A more in-depth discussion comparing the attenuated vaccine and the mRNA vaccine should be included.

Comments on the Quality of English Language

The language needs significant enhancement; it should employ a more formal and comprehensible style of scientific English.

Author Response

Dear reviewer,

Thank you for your valuable comments. I have revised the article according to your comments and those of other reviewers.

Please see the attachment for details.

Reviewer 2 Report

Comments and Suggestions for Authors

This paper is considered to be a very interesting paper as it demonstrates in a mouse model the application of an mRNA vaccine platform with new potential to develop a vaccine against PRRSV, for which no effective vaccine has yet been developed.

Please see below for suggestions for major and minor corrections.

1.     In PRRSV, the heterodimer structure of GP5 and M and the heterotrimer structure of GP2-3-4 have been known to be important for the formation of neutralizing antibodies or vaccine efficacy. The reason for using GP5 alone or GP2-5-M specifically should be explained in the introduction or M&M.

2.     Information on purchasing antibodies for T cell analysis is missing in M&M.

3.     The size and low clarity of Figure 2 make it very difficult to see. Correction is needed.

4.     The authors measured the activation of CD4 or CD8 T cells expressing TNF-alpha. Are these T cell populations that have previously been classified? It is questionable whether the function of these cells has been proven.

5.     The left picture in Figure 10 is difficult to see and its contents cannot be confirmed, so it is recommended to delete it.

6.     As a result, the GP5 only mRNA vaccine induced the highest level of neutralizing antibodies, which contradicts the results of previous studies showing that maintaining the existing GP5-M heterodimer structure is very important. Discussion on this issue should be reflected in the discussion.

7.     In my personal opinion, I believe that many parts that should be discussed in the discussion are missing. I hope that the discussion includes detailed discussion of the results of each development and evaluation stage.

8.     In particular, the results of the two neutralizing antibody test methods are significantly different, and the reason for these differences and the analysis of the results should be included in the discussion.

9.     Since the results were only proven in mice, it is believed that it should be evaluated in pigs, the actual target animal. It would be very helpful if you could discuss some of the expected difficulties and challenges that arise during these evaluations in pigs.

Author Response

(The authors gave the same response as above.)

Reviewer 3 Report

Comments and Suggestions for Authors

In the manuscript of Zhuo et al, the authors describe construction and testing in mouse model mRNA vaccine expressing a single or fused structural proteins of PRRSV.

1.       Objectives: the objectives and the rationale of the study clearly stated.

2.       Methods: The methods are described in sufficient detail to understand the approach used. There is no description of the statistical tests used in the study.

3.       Results: It is not clear if statistics were applied in the experiments presented in figures 9 and 10.

4.       Interpretation: Conclusions are supported by the obtained data. My comment is that numerous efforts have been made in the past to develop a novel PRRSV vaccine with improved safety and protection efficacy. This includes construction of DNA vaccines, subunit vaccines, peptide vaccine and  virus-vectored vaccines. Many of them expressed gene encoding GP5. However, there is little evidence that these vaccines provide substantial protection against PRRSV outside small experimental studies.

5.       Other comments: English must be improved.

Lines 16 – 36. Abstract must contain summary of study design, objectives, obtained results and conclusions. Minimum of general phrases.

Lines 23, 29 and 387 – the contribution of ADE to PRRSV infection remains controversial, especially in pigs.

Line 51 – The modern taxonomy of PRRSV is as following: order Nidovirales, family Arteriviridae, genus Betaarterivirus.

Line 54 – replace “capsule”  with “envelope”.

Line 55 – pH>7.5 is not acid.

Line 58 – replace “gene grope” with “genome”.

Line 60 – remove “GP” in “GP5a”, since this is not a glycoprotein.

Line 85 – replace “good immune efficacy” with “good immunogenicity in mice”.

Figure 1 is not referenced in the text of the manuscript. Also, this figure can be omitted.

Figure 6 – Show sizes of proteins in the picture. Correct typo “Nagetive”.

Figure 7 – Properly label the figure panels.

Figures 8 and 9 – Sera were collected on days 14 and 28. Clarify what time point was analysed and presented in these figures.

Line 290 – this is GP5-specific antibodies since plates were coated with the recombinant GP5 protein.

Lines 291 – 298 – can be omitted.

Lines 303 -316 – can be omitted or moved to the Materials and Methods section.

Line 385. Emphasize that these data are obtained in mouse model, and they need to be confirmed in pigs. 

Comments on the Quality of English Language

Sections 2.2.2 through 2.5.3 must be re-written accordingly rules of academic writing. 

Author Response

(The authors gave the same response as above.)

Reviewer 4 Report

Comments and Suggestions for Authors

I reviewed the manuscript entitled “The mRNA vaccine expressing single and fused structural proteins of PRRSV induces significant cellular and humoral immune responses in BalB/C mice”. In this study authors developed a potential vaccine candidate against PRRSV, that was tested mice.

Overall, I consider that this development might be a potential value for the protection of pigs against PRRSV, however, this potential candidate vaccine was not tested in pigs the natural host of this disease. In fact, there are studies indicating that rodents are not an adequate model for PRRSV. This situation has to be mentioned and explained in the discussion as a main limitation of this study.

Rosenfeld P, Turner PV, MacInnes JI, Nagy E, Yoo D. Evaluation of porcine reproductive and respiratory syndrome virus replication in laboratory rodents. Can J Vet Res. 2009 Oct;73(4):313-8. PMID: 20046635; PMCID: PMC2757714.

Based on the expressed above, I suggest authors to change the title for: A potential mRNA vaccine candidate expressing single and fused structural proteins of PRRSV induces significant cellular and humoral immune responses in BalB/C mice.

Other comments for specific sections:

Abstract: This section should be drastically improved, including more information about the methodology and the results obtained in this study. The current, format of this section looks like a part of the introduction.

Introduction: Please, include more information about the problems with the use of current vaccines, highlight the role of recombination in the emergence of new strains.

Methods: Figure 1 should be improved to express information presented in section 2.1.

Results: In Figure 8 Y axis should be OD value instead of neutralizing antibodies as expressed in the legend. This ELISA is detecting total antibodies not just neutralizing. Also, confirm if in the experiments presented in figures 8 and 9 the same PRRSV vaccine was used. It is interesting that in figure 8 authors show that one of the developments can induce higher levels of antibodies than the vaccine (positive control). However, in figure 9, vaccine is much superior inducing neutralizing antibodies. Based on these results (Figure 9) explain how this mRNA candidate vaccine may perform better in pigs than the current PRRSV vaccines. It highlights again the lack of testing in pigs.

Author Response

(The authors gave the same response as above.)

Round 2

Reviewer 3 Report

Comments and Suggestions for Authors

lines 45-47 Only PRRSV belongs to the genus β-arterivirus. Delete SHFV, LDV and EAV. 

line 49 - Replace the text with "Infectious viral particles are rapidly inactivated by high and low pH and high temperatures..." 

Change the title of Figure 8 to "GP5-specific ELISA tiers in serum."

Comments on the Quality of English Language

English edition is required.

Author Response

Dear reviewers,

I have revised the article in accordance with your requirements and suggestions, and you can see these changes in the attachment.

Reviewer 4 Report

Comments and Suggestions for Authors

I like to thank the authors for their responses to my questions. At this point, I don't have more concerns about this manuscript. 

Author Response

Dear reviewers,

I'm very glad to hear what you said. With your help, this article has been greatly improved. Thank you very much for your suggestions and comments.